# Awareness of Appropriate Antibiotic Use in Primary Care for Influenza-Like Illness: Evidence of Improvement from UK Population-Based Surveys

**DOI:** 10.3390/antibiotics9100690

**Published:** 2020-10-13

**Authors:** Koen B. Pouwels, Laurence S. J. Roope, James Buchanan, Liz Morrell, Sarah Tonkin-Crine, Michele Peters, Leah F. Jones, Enrique Castro-Sánchez, Derrick W. Crook, Tim Peto, Christopher C. Butler, Julie V. Robotham, A. Sarah Walker, Sarah Wordsworth

**Affiliations:** 1Health Economics Research Centre, Nuffield Department of Population Health, University of Oxford, Oxford OX3 7LF, UK; koen.pouwels@ndph.ox.ac.uk (K.B.P.); james.buchanan@dph.ox.ac.uk (J.B.); liz.morrell@ndph.ox.ac.uk (L.M.); sarah.wordsworth@dph.ox.ac.uk (S.W.); 2NIHR Health Protection Research Unit in Healthcare Associated Infections and Antimicrobial Resistance at University of Oxford in Partnership with Public Health England, Oxford OX1 2JD, UK; sarah.tonkin-crine@phc.ox.ac.uk (S.T.-C.); derrick.crook@ndcls.ox.ac.uk (D.W.C.); tim.peto@ndm.ox.ac.uk (T.P.); christopher.butler@phc.ox.ac.uk (C.C.B.); julie.robotham@phe.gov.uk (J.V.R.); sarah.walker@ndm.ox.ac.uk (A.S.W.); 3NIHR Oxford Biomedical Research Centre, John Radcliffe Hospital, University of Oxford, Oxford OX3 9DU, UK; 4Nuffield Department of Primary Care Health Sciences, University of Oxford, Oxford OX2 6GG, UK; 5Health Services Research Unit, Nuffield Department of Population Health, University of Oxford, Oxford OX3 7LF, UK; Michele.peters@ndph.ox.ac.uk; 6Primary Care and Interventions Unit, Public Health England, Gloucester GL1 1DQ, UK; leah.jones@phe.gov.uk; 7School of Health Sciences, Division of Nursing, University of London, London EC1V 0HB, UK; Enrique.castro-sanchez@city.ac.uk; 8Nuffield Department of Medicine, John Radcliffe Hospital, University of Oxford, Oxford OX3 9DU, UK; 9Oxford University Hospitals NHS Trust, Oxford OX3 9DU, UK; 10Modelling and Economics Unit, National Infection Service, Public Health England, London NW9 5EQ, UK

**Keywords:** antibiotics, antimicrobial resistance, survey

## Abstract

Influenza-like illnesses (ILI) account for a significant portion of inappropriate antibiotic use. Patient expectations for antibiotics for ILI are likely to play a substantial role in ‘unnecessary’ antibiotic consumption. This study aimed to investigate trends in awareness of appropriate antibiotic use and antimicrobial resistance (AMR). Three sequential online surveys of independent representative samples of adults in the United Kingdom investigated expectations for, and consumption of, antibiotics for ILI (May/June 2015 (*n* = 2064); Oct/Nov 2016 (*n* = 4000); Mar 2017 (*n* = 4000)). Respondents were asked whether they thought antibiotics were effective for ILI and about their antibiotic use. Proportions and 95% confidence intervals (CI) were calculated for each question and interactions with respondent characteristics were tested using logistic regression. Over the three surveys, the proportion of respondents who believed antibiotics would “definitely/probably” help an ILI fell from 37% (95% CI 35–39%) to 28% (95% CI 26–29%). Those who would “definitely/probably” visit a doctor in this situation fell from 48% (95% CI 46–50%) to 36% (95% CI 34–37%), while those who would request antibiotics during a consultation fell from 39% (95% CI 37–41%) to 30% (95% CI 29–32%). The percentage of respondents who found the information we provided about AMR “new/surprising” fell from 34% (95% CI 32–36%) to 28% (95% CI 26–31%). Awareness improved more among black, Asian and minority ethnic (BAME) than white people, with little other evidence of differences in improvements between subgroups. Whilst a degree of selection bias is unavoidable in online survey samples, the results suggest that awareness of AMR and appropriate antibiotic use has recently significantly improved in the United Kingdom, according to a wide range of indicators.

## 1. Introduction

Antibiotic use is a key driver of antibiotic resistance [1,2]. In England, around 80% of antibiotics are prescribed in primary care [3]. For most conditions, only a minority of antibiotic prescriptions given are necessary [4]. Moreover, a large part of the variation in prescribing rates between general practices cannot be explained by differences in patient comorbidities or other factors such as smoking [5,6,7]. Almost half of all antibiotics prescribed in primary care are for respiratory tract infections (RTIs) such as influenza-like illnesses (ILI), which do not generally require antibiotics [8]. General practitioners (GPs) are more likely to prescribe antibiotics when patients request them or are perceived to want them [9,10]. Thus, while there are many reasons why GPs might prescribe unnecessary antibiotics for ILI [11], patients’ attitudes towards antibiotics and AMR, and patient consultations and antibiotic requests for ILI may play an important role. 

Prior to the timeframe for our study (May 2015–March 2017), a large proportion of the United Kingdom (UK) population believed that antibiotics were effective for ILI, about one third saying they would expect their GP to prescribe antibiotics [12]. Using three sequential online surveys, originally designed to evaluate what drives such beliefs and how different fear-based messages could modify such beliefs [13,14], we assess here potential changes over time in a range of indicators of awareness of AMR and appropriate antibiotic use in the overall population and in subgroups. 

## 2. Methods

### 2.1. Survey Design

Three web-based surveys were conducted in which members of the general public were asked questions about antibiotics and AMR [13,14]. The surveys are described in detail elsewhere [13,14].

The 1st survey [13], conducted in May–June 2015, was primarily designed to investigate what drives patient expectations for antibiotics for ILI; particularly whether AMR-awareness, risk preference or time preference play a role. The 2nd and 3rd surveys [14], conducted during Oct–Nov 2016 and March 2017, respectively, were primarily designed to develop and test the impact of providing different types of information about antibiotic use and AMR on consultations and requests for antibiotics for ILI.

Here we focused on 9 key questions and corresponding variables that were common to all 3 surveys and thus permitted analyses of trends over time in awareness of AMR and attitudes towards the use of antibiotics for ILIs.

The surveys (see Appendix A) asked respondents to imagine Health State A. This health state was intended to describe symptoms of ILI, for which antibiotics are generally not necessary [4], but for which many members of the general public believe antibiotics are effective [12].

### 2.2. Health State A: You Have


*A temperature,*

*Aching muscles,*

*A headache,*

*A dry chesty cough,*

*A sore throat,*

*And you feel weak*


Health state A was described symptomatically because individuals are likely to interpret terms like “flu” or “virus” differently. Moreover, some respondents may know that antibiotics are not required for viral conditions but not realise that these symptoms are more consistent with a viral infection. Respondents were asked how they would respond in the hypothetical situation of having experienced Health State A for 5 days. They were asked whether they thought they would go to see a GP about these symptoms. Except for those who said, ‘Definitely not’, respondents were then asked whether if they did see a GP, they thought they would ask for antibiotics. Respondents were also asked whether they thought antibiotics would be likely to help these symptoms and whether they had taken antibiotics for similar symptoms in the last 12 months. Participants with dependent children then answered analogous questions regarding their youngest child being ill with ILI symptoms, consulting a GP and requesting antibiotics. 

To assess AMR-awareness, respondents were given the information in Box 1, which paraphrased text on the websites of 4 institutions with initiatives aiming to improve AMR-awareness [15,16,17,18]. The message was the same across surveys, though the wording was simplified slightly after Survey 1 to improve readability and comprehension (Box 1). Participants were asked how “surprising” (Survey 1) or “new” (Surveys 2 and 3) they found the information. 

Together, the questions provided 9 indicators of awareness of AMR and attitudes towards antibiotic use for ILI (Box 2). The surveys also captured information on other factors that might be associated with awareness of AMR and antibiotic use (Table 1).

Box 1Information about antimicrobial resistance (AMR) given to survey respondents.
***Information wording in Survey 1***
  *“Antibiotic resistance occurs when an antibiotic loses its ability to effectively control or kill growing bacteria. It is an increasingly serious threat to public health. Without effective antibiotics, many routine treatments will become increasingly dangerous. Setting broken bones, and even basic operations, rely on access to antibiotics that work. Antibiotic resistance is believed to be caused by unnecessary use of antibiotics, and inappropriate use, such as not taking them as prescribed, skipping doses, or saving them for later use.”*
***Information wording in Surveys 2 and 3***
  *“Antibiotic resistance happens when an antibiotic no longer kills or controls growing bacteria. It is an increasingly serious threat to public health. Without antibiotics that work well, many routine treatments will become increasingly dangerous. Setting broken bones, and even basic operations, rely on access to antibiotics that work. Antibiotic resistance is believed to be caused by unnecessary use of antibiotics, and inappropriate use, such as not taking them as prescribed, skipping doses, or saving them for later use.”*

NOTE: The wording in Surveys 2 and 3 was intended to have exactly the same meaning as that in Survey 1. Minor changes in wording were made to the first and third sentences to make them easier to read and understand.

Box 2Nine indicators of awareness of AMR and attitudes to the use of antibiotics for ILI.
**All Respondents**
  *“At this point [5 days of ILI symptoms], do you think you would go to see a GP about these symptoms?” [Definitely/Probably/Probably not/Definitely not/Don’t know]*  *“If you went to see a GP about these symptoms, do you think you would ask for antibiotics?” [Definitely/Probably/Probably not/Definitely not/Don’t know]*  *“Do you think antibiotics would be likely to help these symptoms?”*  *[Definitely/Probably/Probably not/Definitely not/Don’t know]*  *“To the best of your knowledge, have you taken antibiotics for symptoms similar to these in the last 12 months?” [Yes/No]*  *“To what extent is this information [Text in Box 1] new (surprising) to you?” [Very (surprising) new/Somewhat (surprising) new/Not very (surprising) new/Not at all (surprising) new]*
**Respondents With Children Only**
  *“At this point [5 days of ILI symptoms], do you think you would take your child to see a GP about these symptoms?” [Definitely/Probably/Probably not/Definitely not/Don’t know]*  *“If you did take your child to see a GP about these symptoms, do you think you would ask for antibiotics?” [Definitely/Probably/Probably not/Definitely not/Don’t know]*  *“Do you think antibiotics would be likely to help your child in this situation?” [Definitely/Probably/Probably not/Definitely not/Don’t know]*  *“To the best of your knowledge, has your child taken antibiotics for symptoms similar to these in the last 12 months?” [Yes/No]*

### 2.3. Survey Participants

The surveys were conducted online using panels of respondents provided by Survey Sampling International (SSI). For the first survey, SSI was commissioned to obtain a sample of at least 2000 completed responses, representative of adult members of the general public in terms of gender, age, ethnicity and geographic region. For the second and third surveys, SSI was commissioned to obtain samples of 4000 completed responses, again representative of the UK-resident adult members of the general public (the larger sample sizes needed to support a randomized experiment with three arms reported elsewhere [14]). In total, 6280; 28,887; and 8317 SSI panel members were invited via email for the first, second and third survey, respectively. It was also possible for SSI panel-members to access the surveys via SSI’s website. Respondents were only allowed to participate in one survey, thereby preventing potential learning from previous surveys. 

Respondents were offered a small incentive to complete the surveys in the form of ‘Nectar points’ (a loyalty-card scheme via which customers accrue discounts at outlets including supermarkets and restaurants), worth a total of approximately £0·60.

### 2.4. Statistical Analysis 

Chi-squared tests were used to test whether the 9 indicators of AMR-awareness and attitudes towards antibiotics were the same in Survey 1 (May–June 2015) and Survey 3 (March 2017). Indicators were also reported for Survey 2 (Oct–Nov 2016) for full disclosure, but we did not test for differences between Surveys 1 and 2 or Surveys 2 and 3. This was because attitudes may be different before (Survey 2) and after (Surveys 1 and 3) the seasonal peak in ILI incidence [19]. Confidence intervals were estimated using Wilson score intervals. 

To explore whether awareness/attitudes improved more/less in certain subgroups, we used logistic regression models that allowed for interaction terms between time and respondent characteristics. Because the surveys were not equally spaced in time, and, as above, because attitudes may be different before and after the seasonal peak in ILI incidence [19], we only included data from Surveys 1 and 3 in this analysis and simplified ‘time’ to a dummy variable. The following variables were included as main effects in each model: Gender, age, age of child (for child-related questions), household income, being unemployed, white ethnicity, having completed higher education and having been born in the United Kingdom. Interactions between each variable and time (Survey 1 vs. 3) were introduced separately to each model, only one interaction included each time. We performed complete case analyses as variables like household income are unlikely to be reliably imputed using data from available variables. Moreover, complete case analysis is valid when missingness in a covariate is independent of the outcome conditional on the regression model’s covariates [20,21]. The estimated marginal effects and *p*-values of the interaction terms were plotted for each regression model. All statistical analyses were done using R version 3.5.1. 

## 3. Results

Survey 1 was completed by 2064 respondents. Surveys 2 and 3 were each completed by 4000 respondents (Table 1). 

### 3.1. Shifts in Attitudes over Time

Awareness of AMR and appropriate antibiotic use improved over time (Table 2). While 48% of respondents said that they would go to the GP if they experienced 5 days of ILI symptoms in Survey 1 (May–June 2015), only 36% of respondents indicated they would do so in Survey 3 (March 2017). In Survey 1, 39% of respondents indicated that if they did see a GP for ILI symptoms, they would ask the GP for antibiotics. By Survey 3, this percentage had decreased to 30%. There was a substantial overlap between those respondents who would go to the GP for ILI symptoms and those who would ask for antibiotics. In Survey 1, of the 706 respondents who indicated that they would ask for antibiotics if they went to see a GP for ILI symptoms, 561 (79%) said that they would indeed be likely to see a GP in this situation. In Survey 3, of the 975 respondents who indicated that they would ask for antibiotics if they went to see a GP, 736 (75%) said that they would be likely to see a GP in this situation.

In total, 37% of respondents from Survey 1 thought that antibiotics would help if they experienced ILI, and 53% thought antibiotics would be effective for their child with ILI. By Survey 3, these figures had decreased to 28% and 43%, respectively. The percentage of respondents who indicated that the AMR information (Box 1) was new/surprising declined from 34% in Survey 1 to 28% in Survey 3. 

The percentage of respondents who had used antibiotics for ILI in the 12 months before the survey also declined over time, from 21% in Survey 1 to 14% in Survey 3. Although past antibiotic use for ILI was higher in children across the surveys, a similar decline in use was observed between Surveys 1 and 3 from 25% to 20%. 

### 3.2. Shifts in Awareness among Subgroups

For most indicators, there was little evidence of subgroups for which awareness improved less than for others (Figure 1 and Appendix A). The only factor for which there was evidence (*p* < 0.05 for interaction with time) for multiple indicators that improvement in awareness differed by subgroup was ethnicity. Awareness improved more among BAMEs than whites for the following three indicators: (1) Going to GP with child with ILI, (2) asking for antibiotics when going to GP with ILI, and (3) finding the AMR information that was provided new/surprising. Effectively, awareness among BAMEs and whites became more similar over time for these indicators. 

## 4. Discussion 

### 4.1. Summary

The findings from these surveys suggest that awareness of AMR and appropriate antibiotic use have improved over time. In the most recent survey, respondents were substantially more likely to be aware of AMR and appropriate antibiotic use. Moreover, the number of adults, and their children, who had taken antibiotics for ILI in the previous 12 months declined. 

Just as diseases often cluster within certain populations according to sociodemographic factors [22], so too does antibiotic use. There is evidence in England and Wales of significant variation in antibiotic prescribing levels by area-level deprivation, with higher prescribing levels in more deprived areas [7,23,24]. Moreover, this association is not fully explained by differences in the prevalence of common chronic conditions or smoking status [7,24]. In previous work using the present dataset, several of the indicators used in this study have been shown to be associated with a number of sociodemographic characteristics [13]. In multivariable regression analyses, there were generally strong independent associations with gender and ethnicity and some evidence of independent associations with age and self-reported health status [13]. Interestingly, after adjusting for other characteristics, there was only limited evidence of independent association with household income [13]. 

Encouragingly, in this study, we found generally little evidence that the observed trends in improving awareness were restricted to or were weaker in any particular population subgroups. The exception was ethnicity, where some AMR and appropriate antibiotic use awareness indicators improved more among BAMEs. However, this is also encouraging as the indicators in Survey 1 suggest that awareness of AMR and appropriate antibiotic use was initially poorer among BAMEs than whites [13]. Nevertheless, despite the observed improvements, in the most recent survey, a substantial proportion of respondents still said that the information we gave to them in the survey about AMR was new to them and that antibiotics would be likely to help ILI symptoms. 

### 4.2. Strengths and Limitations

A key limitation of this study is that only members of the online survey panels participated. A degree of selection bias is unavoidable in online survey samples as persons with no interest in completing surveys, no internet-access or no basic computer literacy cannot be included. Nevertheless, the panels were broadly representative of the general UK population in terms of age, gender, ethnicity, region and unemployment, though, as is common in online surveys, the percentage with higher education was relatively high [13]. 

Respondents’ answers may not reflect what they will actually do in practice. However, it seems likely that the observed changes in expectations and attitudes are likely to support desirable consultation behaviour in future.

The surveys were not explicitly designed to investigate changes in attitudes over time and there was limited scope to include further questions that might have helped with interpreting the results. For example, additional questions might have sought to ascertain awareness of media campaigns or news about AMR around the time the surveys were in the field. The three surveys were conducted during a period in which the need to tackle AMR was gaining increasing prominence in the UK. In particular, the Review on Antimicrobial Resistance, commissioned by the UK government in July 2014, produced an initial report [25] in December 2014 and its final report and recommendations [26] in the summer of 2016. Stark warnings from this review, including estimates that AMR could result in up to 10 million deaths by 2050, generated considerable media attention. Other notable public events included European Antibiotic Awareness Day, which occurs annually on 18th November, and the ongoing Antibiotic Guardian Campaign [16], which launched in September 2014 and invites people to make a pledge related to what they personally will do to help to conserve antibiotics. There is evidence that the Antibiotic Guardian Campaign has been an effective tool for engaging people with the problem of AMR, increasing self-reported knowledge and changing behaviour, especially among those with prior awareness of the topic [27]. There were also other interventions such as the Treat Antibiotics Responsibly, Guidance, Education, Tools (TARGET) toolkit [28,29], which was designed to help influence prescribers’ and patients’ personal attitudes, social norms and perceived barriers to optimal antibiotic prescribing [30]. A pragmatic randomised controlled trial found that interactive workshops, delivered as part of the TARGET intervention, helped to reduce antibiotic dispensing [31]. However, the three surveys from the current study cannot be used to disentangle to what extent improvements in awareness of AMR and appropriate use of antibiotics were driven by public campaigns or interventions such as these. The potential relative and incremental benefit of public campaigns and other interventions, including their possibly differing impacts in different population subgroups, would ideally be assessed via randomised designs [14,32].

The three surveys differed with respect to their timing in relation to the seasonal peak in ILI incidence. Moreover, the severity of the influenza seasons during the study period also varied [33]. Attitudes may have been influenced by the proximity to and the severity of these seasons. However, it is not feasible to disentangle these possible effects using these data. 

The setting for this study was the UK, and a further limitation is that trends in awareness of appropriate antibiotic use are likely to vary across countries. For example, a meta-analysis of cross-sectional studies published during 2000 to 2013, stratified by continents, found that an inappropriate answer to the statement “Antibiotics are useful for cold and flu,” was given by 69.0% of respondents in Oceania but only 28.0% in Europe [34].

### 4.3. Comparison with Existing Literature

During the study period, antibiotic prescribing in primary care in England declined by approximately 8% between the respective second quarters of 2015 and 2017 [35]. This decline is likely largely due to a reduction in antibiotic use for RTIs, such as ILI, that generally do not require antibiotics [4,5]. Our results suggest that the reduction in prescriptions may not only be a consequence of behaviour change among GPs but may also be partly due to simultaneous changes in the attitudes and awareness of patients. 

While changes in attitudes may lead to fewer antibiotics being prescribed, reducing antibiotic prescribing has also been shown to reduce GP consultation rates for both upper and lower RTIs [36,37]. This is particularly relevant given that consultation rates explain a large part of the variation in antibiotic prescribing [6]. It is, therefore, encouraging that we not only found that patients were less likely to ask for antibiotics for ILI, but also less likely to consult, in the most recent survey. 

Public perceptions of AMR and appropriate use of antibiotics in primary care have been surveyed in previous studies [12,38,39,40]. Our descriptive results broadly concur with previous studies in confirming wide-ranging misunderstanding about how effective antibiotics are for ILI, and lack of awareness of AMR. 48% of our respondents from Survey 1 (May–June 2015) said they would ‘definitely/probably’ consult a GP if they had an ILI for five days. By comparison, McNulty et al. [12] found that in 2011 only 20% of respondents who had RTI symptoms in the last six months reported contacting or visiting a doctor. However, their question was much broader, including all those who had experienced not only ILI but also sore throat, cold or cough. Our Survey 1 finding, that 39% of respondents would ask a GP for antibiotics for ILI, while 37% believed antibiotics would help, were also broadly consistent with McNulty et al. [12], who reported that 38% of their sample believed antibiotics could kill viruses, while 32% would expect their GP to prescribe antibiotics for ILI. In Survey 1, 21% (39%) of respondents reported having taken antibiotics for ILI (all conditions) in the last 12 months. A study using The Health Improvement Network (THIN) UK primary care database found that approximately 30% of patients were prescribed at least one antibiotic per year for all conditions during 2011–2013 [41]. Therefore, respondents to our survey may have been prescribed antibiotics more frequently than the general UK public. While this may mean that our results are not necessarily generalizable to individuals who use less antibiotics, one could argue that in the case of ILI this is not an important limitation, as improvements are most needed among those who take antibiotics for conditions that generally do not require them. 

### 4.4. Implications for Future Research and Clinical Practice

Although we found an improvement in awareness of AMR and appropriate antibiotic use, there remains a substantial proportion of the UK general population who think that antibiotics would help ILI symptoms, and who would ask their GP for antibiotics in a consultation. There is an urgent need for further research that evaluates how such beliefs might be successfully challenged, for example, in public health campaigns. One promising avenue shown to have the potential for improving the indicators in our surveys is to provide people with fear-based messages about AMR together with empowering information on how to effectively self-manage symptoms without antibiotics [14].

## Figures and Tables

**Figure 1 antibiotics-09-00690-f001:**
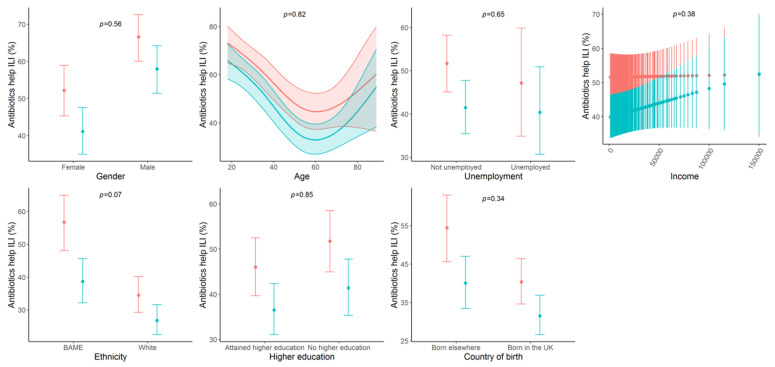
Changes from Survey 1 (red) to Survey 3 (blue) in percentage of individuals that think antibiotics help by population subgroup. NOTES: 1. Marginal averaged percentages and 95% confidence intervals by population subgroup for thinking antibiotics would help with treating influenza-like illness for survey 1 (May–June 2015; red dots and lines) and survey 3 (March 2017; blue dots and lines). 2. *p*-values are given for the interaction between each characteristic and survey number (time).

**Table 1 antibiotics-09-00690-t001:** Respondent characteristics.

	Survey 1 (May–June 2015) (*n* = 2064) ^a^	Survey 2 (Oct–Nov 2016) (*n* = 4000) ^a^	Survey 3 (Mar 2017) (*n* = 4000) ^a^
**Gender**			
Male	994 (48.2%)	1941 (48.5%)	1941 (48.5%)
Female	1067 (51.7%)	2055 (51.4%)	2055 (51.4%)
Missing	3 (0.1%)	4 (0.1%)	4 (0.1%)
**Age (years)**			
Mean (SD)	44.2 (15.7)	46.6 (16.9)	46.5 (16.8)
**Ethnicity**			
White	1821 (88.2%)	3624 (90.6%)	3606 (90.2%)
Black, and minority ethnic (BAME)	221 (10.7%)	352 (8.8%)	372 (9.3%)
Missing	22 (1.1%)	24 (0.6%)	22 (0.6%)
**Religion**			
Christian	1019 (49.4%)	1986 (49.6%)	1858 (46.5%)
No or other religion	988 (47.9%)	1913 (47.8%)	2049 (51.2%)
Missing	57 (2.8%)	101 (2.5%)	93 (2.3%)
**Married/civil partnership/live with a partner**			
Yes	1351 (65.5%)	2720 (68.0%)	2668 (66.7%)
No	713 (34.5%)	1280 (32.0%)	1332 (33.3%)
**Have dependent children**			
Yes	816 (39.5%)	1600 (40%)	1600 (40%)
No	1248 (60.5%)	2400 (60%)	2400 (60%)
**Higher education**			
Attained higher education	954 (46.2%)	1875 (46.9%)	1756 (43.9%)
Did not attain higher education	1093 (53.0%)	2125 (53.1%)	2244 (56.1%)
Missing	17 (0.8%)	0 (0%)	0 (0%)
**Unemployed**			
Yes	105 (5.1%)	202 (5.0%)	184 (4.6%)
No	1959 (94.9%)	3798 (95.0%)	3816 (95.4%)
**Household equivalent income £**			
Mean (SD)	22,405 (18,343)	22,109 (17,123)	20,477 (15,012)
Missing	186 (9.0%)	390 (9.8%)	357 (8.9%)
**Region**			
England	1780 (86.2%)	3381 (84.5%)	8542 (84.9%)
Northern Ireland	32 (1.6%)	104 (2.6%)	240 (2.4%)
Scotland	161 (7.8%)	318 (8.0%)	797 (7.9%)
Wales	91 (4.4%)	197 (4.9%)	485 (4.8%)
**Antibiotic use**			
Taken antibiotics for any condition in last 12 months	815 (39%)	1367 (34%)	1422 (36%)

NOTE: *^a^*. Survey Sampling International was commissioned to provide at least 2000 completed responses in Survey 1 [13]. The larger sample sizes of 4000 were commissioned in Surveys 2 and 3 to support a randomized experiment with three arms [14].

**Table 2 antibiotics-09-00690-t002:** Shifts in attitudes across the three surveys.

	Survey 1 (May–June 2015) *n* = 2064 n (%; 95% CI)	Survey 2 (Oct–Nov 2016) *n* = 4000 n (%; 95% CI)	Survey 3 (March 2017) *n* = 4000 n (%; 95% CI)	*p*-Value ^d^
Definitely/Probably go to GP for ILI	988 (48%; 46–50%)	1587 (40%; 38-–41%)	1438 (36%; 34–37%)	<0.0001
Would ask GP for antibiotics if went	706/1816 ^a^ (39%; 37–41%)	1084/3298 ^a^ (33%; 31–24%)	975/3236 ^a^ (30%; 29–32%)	<0.0001
Think antibiotics would definitely/probably help ILI	762 (37%; 35–39%)	1,153 (29%; 27–30%)	1,112 (28%; 26–29%)	<0.0001
Have taken antibiotics for ILI in last 12 months	426 (21%; 19–22%)	595 (15%; 14–16%)	580 (14%; 13–16%)	<0.0001
Definitely/Probably go to GP for child with ILI	673/816 ^b^ (82%, 80–85%)	1232/1600 ^b^ (77%; 75–79%)	1191/1600 ^b^ (74%; 72–77%)	<0.0001
Would ask GP for antibiotics for child with ILI if went	419/797 ^c^ (53%; 49–56%)	734/1519 ^c^ (48%; 46–51%)	671/1526 ^c^ (44%; 41–46%)	<0.0001
Think antibiotics would definitely/probably help child with ILI	430/816 ^b^ (53%; 49–56%)	740/1600 ^b^ (46%; 44–49%)	688/1600 ^b^ (43%; 41–45%)	<0.0001
Child has taken antibiotics for ILI in last 12 months	200/816 (25%; 22–28%)	338/1600 (21%; 19–23%)	319/1600 (20%; 18–22%)	0.0097
AMR information (Box 1) is very/somewhat new (very/somewhat surprising in Survey 1)	705 (34%; 32–36%)	303/1000 (30%; 28–33%)	285/1000 (28%; 26–31%)	0.0017

NOTES: ^a^. Denominator here is all respondents except those who said they would ‘definitely not’ go to GP for ILI; ^b^. The denominator here is number of respondents with dependent children; ^c^. The denominator here is all respondents except those who said they would ‘definitely not’ go to GP for child with ILI; ^d^. *p*-values compare Survey 1 with Survey 3.

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
