# Peer review of "Awareness of Appropriate Antibiotic Use in Primary Care for Influenza-Like Illness: Evidence of Improvement from UK Population-Based Surveys"

_antibiotics, 2020, doi:10.3390/antibiotics9100690_

Round 1

Reviewer 1 Report

 Does the household income have any effect on attitudes, and can the authors use specific income ranges to make any correlations?

Several studies hsow this trend, for example,

Alvarez-Uria G, Gandra S, Laxminarayan R. Poverty and prevalence of
antimicrobial resistance in invasive isolates. Int J Infect Dis 2016;52:59–
61.

Kirby A, Herbert A. Correlations between income inequality and
antimicrobial resistance. PLoS One 2013;8:e73115.

Nomamiukor B, Horner C, Kirby A, et al. Living conditions are
associated with increased antibiotic resistance in community isolates of
Escherichia coli. J Antimicrob Chemother 2015;70:3154–8.

The authors should also expand the discussion in view of other studies, and try to discuss any disease burden/social factors that may be derived from the current datasets.  

Singer M, Bulled N, Ostrach B, et al. Syndemics and the biosocial
conception of health. Lancet 2017;389:941–50.

Author Response

We are grateful to the reviewer for providing some excellent suggestions and we believe the revised paper has improved in response to these. We have replied below to each of these suggestions.

We have tracked changes in the revised manuscript and also provided a clean, untracked version. Line numbers given throughout our response refer to the version with tracked changes.

Reviewer reports:

Reviewer #1:

Does the household income have any effect on attitudes, and can the authors use specific income ranges to make any correlations?

Several studies hsow this trend, for example,

Alvarez-Uria G, Gandra S, Laxminarayan R. Poverty and prevalence of
antimicrobial resistance in invasive isolates. Int J Infect Dis 2016;52:59–
61.

Kirby A, Herbert A. Correlations between income inequality and
antimicrobial resistance. PLoS One 2013;8:e73115.

Nomamiukor B, Horner C, Kirby A, et al. Living conditions are
associated with increased antibiotic resistance in community isolates of
Escherichia coli. J Antimicrob Chemother 2015;70:3154–8.

RESPONSE: We are grateful to the reviewer for giving us the opportunity to clarify this point. The associations between household income and attitudes indicators in the dataset have been extensively investigated in our earlier work [Roope et al., 2018 (reference 13 in manuscript)]. Interestingly, in multivariable regression analyses, after adjusting for other characteristics there was generally little evidence of an independent association between household income and the attitude indicators. However, as discussed in response to the reviewer’s following point, there is evidence of associations with several other socio-demographic indicators. As discussed further below, we have added a substantial paragraph (lines 201-211 in revised manuscript) discussing the associations between socio-demographic indicators and attitudes. This includes the following text specifically regarding income (lines 209-211): “Interestingly, after adjusting for other characteristics, there was only limited evidence of independent association with household income [Roope et al., 2018].”

The authors should also expand the discussion in view of other studies, and try to discuss any disease burden/social factors that may be derived from the current datasets.  

Singer M, Bulled N, Ostrach B, et al. Syndemics and the biosocial
conception of health. Lancet 2017;389:941–50.

RESPONSE: We are grateful to the reviewer for the suggestion to expand the discussion in view of other studies (including the Singer et al., 2017 study). We especially welcome the opportunity to clarify some notable associations between socio-demographic indicators (including health) and attitudes indicators. As noted above, the associations between a wide range of socio-demographic indicators, with the attitudes indicators in the dataset have been extensively investigated in our earlier work (Roope et al., 2018 (reference 13 in manuscript). We have added the following paragraph (lines 201-211 in the revised manuscript) discussing the associations between socio-demographic indicators and attitudes:

“Just as diseases often cluster within certain populations according to sociodemographic factors [Singer et al., 2017)], so too does antibiotic use. There is evidence in England and Wales of significant variation in antibiotic prescribing levels by area-level deprivation, with higher prescribing levels in more deprived areas [Hope et al., 2020; Thomson et al., 2020; Adekanmbi et al., 2020]. Moreover, this association is not fully explained by differences in the prevalence of common chronic conditions or smoking status [Hope et al., 2020; Adekanmbi et al., 2020]. In previous work using the present dataset, several of the indicators used in this study have been shown to be associated with a number of sociodemographic characteristics [Roope et al, 2018]. In multivariable regression analyses, there were generally strong independent associations with gender and ethnicity and some evidence of independent associations with age and self-reported health status [Roope et al, 2018]. Interestingly, after adjusting for other characteristics, there was only limited evidence of independent association with household income [Roope et al, 2018].”

References

-Roope LSJ, Tonkin-Crine S, Butler CC, Crook D, Peto T, Peters M, Walker AS, Wordsworth S. Reducing demand for antibiotic prescriptions: evidence from an online survey of the general public on the interaction between preferences, beliefs and information, United Kingdom, 2015. Eurosurveillance. 2018 Jun 21;23(25):1700424.

-Hope EC, Crump RE, Hollingsworth TD, Smieszek T, Robotham JV, Pouwels KB. Identifying English practices that are high antibiotic prescribers accounting for comorbidities and other legitimate medical reasons for variation. EClinicalMedicine 2018;6:36-41.

-Thomson, K., Berry, R., Robinson, T. et al. An examination of trends in antibiotic prescribing in primary care and the association with area-level deprivation in England. BMC Public Health 20, 1148 (2020). https://doi.org/10.1186/s12889-020-09227-x

-Adekanmbi V, Jones H, Farewell D, Francis NA. Antibiotic use and deprivation: an analysis of Welsh primary care antibiotic prescribing data by socioeconomic status. Journal of Antimicrobial Chemotherapy. 2020 May 25.

Reviewer 2 Report

This study was well prepared while more in-depth writing would be required.

Major concerns:

  1. Have there been the mass-media campaign and announcement of antibiotic use and AMR in this survey areas? More describe and discuss them.
  2. How was the socially epidemic condition of influenza and common cold in this survey area? Can it affect the survey results?
  3. How was the awareness of antibiotic use in other countries? More describe and discuss them.
  4. Do people change the visit to GP with the change of antibiotic prescription? More describe and discuss them.
  5. For instance, how was the change of marketing survey of sales of antibiotic use in the survey area? Do the authors have any additional data for reinforcement and supplement of the survey data?
  6. In this online survey, how were the ethics approved and was informed consent provided?

Author Response

We are grateful to the reviewer for providing some excellent suggestions and we believe the revised paper has improved in response to these. We have replied below to each of these suggestions.

We have tracked changes in the revised manuscript and also provided a clean, untracked version. Line numbers given throughout our response refer to the version with tracked changes.

Reviewer reports:

Reviewer #2:

This study was well prepared while more in-depth writing would be required.

Major concerns:

  1. Have there been the mass-media campaign and announcement of antibiotic use and AMR in this survey areas? More describe and discuss them.

RESPONSE: We are grateful to the reviewer for the suggestion to provide more detail on prominent public campaigns and announcements about AMR. We have added the following text (on lines 240-248) to address this.

“The three surveys were conducted during a period in which the need to tackle AMR was gaining increasing prominence in the UK. In particular, the Review on Antimicrobial Resistance, commissioned by the UK government in July 2014, produced an initial report [Review on Antimicrobial Resistance, 2014] in December 2014 and its final report and recommendations [Review on Antimicrobial Resistance, 2016] in the summer of 2016. Stark warnings from this review, including estimates that AMR could result in up to 10 million deaths by 2050, generated considerable media attention. Other notable public events included European Antibiotic Awareness Day, which occurs annually on 18th November, and the ongoing Antibiotic Guardian Campaign [Antibiotic Guardian Campaign], which launched in September 2014 and invites people to make a pledge related to what they personally will do to help to conserve antibiotics.”

References

-The Review on Antimicrobial Resistance. Antimicrobial resistance: tackling a crisis for the health and wealth of nations. London: UK Government; 2014. Available from: https://amr-review.org/sites/default/files/AMR%20Review%20Paper%20-%20Tackling%20a%20crisis%20for%20the%20health%20and%20wealth%20of%20nations_1.pdf

-The Review on Antimicrobial Resistance. Tackling Drug-Resistant Infections Globally: Final Report and Recommendations. London: UK Government; 2016. Available from: https://amr-review.org/sites/default/files/160525_Final%20paper_with%20cover.pdf

- Antibiotic Guardian Campaign. https://antibioticguardian.com/

  1. How was the socially epidemic condition of influenza and common cold in this survey area? Can it affect the survey results?

However, we agree with the reviewer that this issue deserves more details and discussion. As noted, the three surveys differed with respect to their timing in connection with the seasonal peak in ILI incidence. Moreover, the relevant ILI seasons during the study period also varied in terms of their severity: Survey 1 in May-June 2015 was after the seasonal peak in a severe season with 28,330 deaths [Public Health England, 2019]. Survey 3 in March 2017 was also after, but not so long after, the seasonal peak in a more moderate season with 18,009 deaths [Public Health England, 2019]. Survey 2 in Oct-Nov 2016 was before the peak in the 2016-2017 season and long after the relatively mild 2015-2016 season with 11,875 deaths [Public Health England, 2019]. Attitudes may have been influenced by the proximity to and the severity of these seasons. However, it is not feasible to disentangle these possible effects using these data.

“The three surveys differed with respect to their timing in relation to the seasonal peak in ILI incidence. Moreover, the severity of the influenza seasons during the study period also varied [Public Health England, 2019]. Attitudes may have been influenced by the proximity to and the severity of these seasons. However, it is not feasible to disentangle these possible effects using these data.”

Reference

-Public Health England, 2019. Surveillance of influenza and other respiratory viruses in the UK: Winter 2018 to 2019. Available at 

https://assets.publishing.service.gov.uk/government/uploads/system/uploads/attachment_data/file/839350/Surveillance_of_influenza_and_other_respiratory_viruses_in_the_UK_2018_to_2019-FINAL.pdf

  1. How was the awareness of antibiotic use in other countries? More describe and discuss them.

RESPONSE: We are grateful to the reviewer for asking this, especially as it is important to be clear in the paper that the results apply to the UK setting and that trends in other countries are likely to be different. We have added the following text in the ‘strengths and limitations’ section of the Discussion (lines 261-265).

Reference

-Gualano MR, Gili R, Scaioli G, Bert F, Siliquini R. General population's knowledge and attitudes about antibiotics: a systematic review and meta‐analysis. Pharmacoepidemiology and drug safety. 2015 Jan;24(1):2-10.

  1. Do people change the visit to GP with the change of antibiotic prescription? More describe and discuss them.

RESPONSE: We agree with the reviewer that it is of some interest to note whether the respondents who said they would be likely to visit the GP with ILI symptoms are largely the same respondents as those who said they would ask for antibiotics if they did go. In the revised manuscript, after describing the results for these two indicators, we now outline the relationship between them. We have added the following text (lines 160-166):

There was substantial overlap between those respondents who would go to the GP for ILI symptoms and those who would ask for antibiotics. In Survey 1, of the 706 respondents who indicated that they would ask for antibiotics if they went to see a GP for ILI symptoms, 561 (79%) said that they would indeed be likely to see a GP in this situation. In Survey 3, of the 975 respondents who indicated that they would ask for antibiotics if they went to see a GP, 736 (75%) said that they would be likely to see a GP in this situation.”

  1. For instance, how was the change of marketing survey of sales of antibiotic use in the survey area? Do the authors have any additional data for reinforcement and supplement of the survey data?

RESPONSE: We share the reviewer’s interest in comparing the trends in awareness of appropriate antibiotic use investigated in this study with additional data on actual antibiotic use. The best available data on trends in actual antibiotic use is Public Health England’s ‘Fingertips’ prescribing data. Unfortunately it is not possible to disaggregate this data by indication, but we can look at total antibiotic use in primary care, which declined during the study period. Moreover, there is good evidence that this decline is likely due to a reduction in antibiotic use for RTIs, such as ILI, that generally do not require antibiotics. In fact, we have made this comparison in the manuscript. On lines 268-270 in the revised manuscript we have the text:

During the study period antibiotic prescribing in primary care in England declined by approximately 8% between the respective second quarters of 2015 and 2017 [Public Health England, Fingertips]. This decline is likely largely due to a reduction in antibiotic use for RTIs, such as ILI, that generally do not require antibiotics [Pouwels et al., 2018; Smieszek et al., 2018].

References

-Public Health England. Fingertips. AMR local indicators. Antibiotic Prescribing. Trends. https://fingertips.phe.org.uk/profile/amr-local-indicators/data#page/4/gid/1938132909/pat/152/par/E38000001/ati/7/are/B83620

-Pouwels KB, Dolk FCK, Smith DRM, Robotham J V, Smieszek T. Actual versus ‘ideal’ antibiotic prescribing for common conditions in English primary care. J Antimicrob Chemother 2018; 73: 19–26.

-Smieszek T, Pouwels KB, Dolk FCK, et al. Potential for reducing inappropriate antibiotic prescribing in English primary care. J Antimicrob Chemother 2018; 73: ii36-ii43.

  1. In this online survey, how were the ethics approved and was informed consent provided?

RESPONSE: As we tried to make clear in the manuscript (e.g., lines 55-57), the three surveys analysed in this study were originally designed to support two previous studies that have already been published (Roope et al., 2018 and Roope et al., 2020]). In the ethical approval section in this study, we therefore felt the most appropriate thing to do was to refer the reader to those studies for details of the ethical approval – specifically the Roope et al., 2020 study which contains full details of ethical approval for all three surveys. In the manuscript (lines 342-343) the text therefore reads:

“This study is based on data from three surveys, previously reported by Roope et al. (2018, 2020). Full details of ethical approval for all three surveys are reported in Roope et al. (2020).”

The full ethical approval details for the three surveys reported in Roope et al. (2020) reads as follows – again, we feel it is more appropriate to simply refer to this in the ethics statement but are happy to add a statement closely based on this full text if you would prefer:

“This study is based on data from three surveys. The first survey data existed prior to this study, having been developed separately [Roope et al., 2015], on attitudes to antibiotics and awareness of AMR. Respondents had volunteered to receive information about such surveys from SSI. Completion of the survey was considered as indicating consent, and respondents were able to refuse to participate in the survey at any stage in the process. The third survey (second wave of this study), which is identical to the second survey (first wave of this study), was approved by the University of Oxford’s Central University Research Ethics Committee—reference number R49463/RE001. Following approval of the third survey, the original and second survey were retrospectively considered by the same committee (reference number R57213/RE001) who deemed that ‘it is probable that, had the documentation and a full application been submitted for ethical opinion at the correct time, the committee would have granted ethical approval (perhaps subject to some amendments of the documentation).’ Respondents gave written informed consent before taking part.”

References

-Roope LSJ, Tonkin-Crine S, Butler CC, et al. Reducing demand for antibiotic prescriptions: evidence from an online survey of the general public on the interaction between preferences, beliefs and information, United Kingdom, 2015. Euro Surveill 2018; 23. DOI:10.2807/1560-7917.ES.2018.23.25.1700424.

-Roope LSJ, Tonkin-Crine S, Herd N, et al. Reducing expectations for antibiotics in primary care: a randomised experiment to test the response to fear based messages about antimicrobial resistance. BMC Medicine 2020; 18: 1-11.

Round 2

Reviewer 2 Report

The revised version has been much improved. The authors described the detail on public campaigns and announcements about AMR. If possible, could the authors add the public health impact of these campaigns and announcements among general people, using any data? Or discuss more their influence on the survey.
